



# Steering recoverable autonomous sonde (RECAS) for accessing and studying subglacial lakes

Mikhail A. Sysoev[1], Pavel G. Talalay[1,2], Xiaopeng Fan[1], Nan Zhang[1], Da Gong[1], Yang Yang[1], Ting Wang[1], Zhipeng Deng[1]

[1]Institute of Polar Science and Engineering, Jilin University, Changchun, China

[2]School of Engineering and Technology, China University of Geosciences, Beijing, China

Authors for correspondence: Pavel Talalay, E-mail: ptalalay@yahoo.com; Da Gong, E-mail: gongda@jlu.edu.cn

## Abstract

The study of subglacial lakes requires clean access and sampling technologies. One of the most promising alternatives is the newly developed hot-point RECoverable Autonomous Sonde (RECAS), which allows downward and upward ice drilling and subglacial water sampling while the subglacial lake remains isolated from the surface. The original sonde descends downward under the force of gravity, and the borehole trajectory cannot be controlled. However, in certain cases, the sonde would preferably be able to drill at specific angles and directions, enabling it to follow a desired trajectory (e.g., maintaining verticality within the desired range) or bypass obstacles in the ice (e.g., stones and other inclusions). The general principle for the steering RECAS is to adjust the voltage for the electric thermal head heaters, which provides an opportunity to control the heat distribution on the drill head surface, thereby altering borehole trajectory during drilling. In this paper, the general principles of steering RECAS are described, and experimental results on deviational ice drilling with a controllable electric thermal head are discussed.

## Keywords

Ice drilling technology; Subglacial lakes; Clean access sampling; Thermal sonde;

Steerable system



## 1. Introduction

It is now widely accepted that subglacial hydrological environments are similar to the water distribution found elsewhere on Earth's surface and comprise a vast network of lakes, rivers, and streams located thousands of metres beneath ice caps, glaciers, and the Antarctic and Greenland ice sheets (Bowling et al., 2019; Siegert et al., 2012). A subglacial lake is considered to be any large body of liquid water existing below an ice mass. The water depth of subglacial lakes varies from a few to several hundred metres. As of 2022, a total of 773 subglacial lakes have been identified, including 675 in Antarctica, 64 in Greenland, two beneath the Devon Ice Cap, six beneath Iceland's ice caps, and 26 in valley glaciers (Livingstone et al., 2022). The ice thickness above subglacial lakes may vary from several tens to thousands of metres.

Subglacial lakes provide unique information regarding paleoclimatic conditions, basal hydrology, biogeochemical fluxes, and geomorphic activity. It is anticipated that subglacial lakes harbour relict microbial species capable of thriving in complete darkness, low nutrient levels, high water pressures, and isolation from the atmosphere (Skidmore, 2011). In-situ investigations should not contaminate these subglacial aquatic systems. Currently, hot-water drilling systems are considered the cleanest method for accessing subglacial lakes. US teams successfully accessed the Whillans and Mercer subglacial lakes on the coastal margin of West Antarctica in early 2013 and during the 2018-2019 season, demonstrating the well-proven effectiveness of this technology (Priscu et al., 2021; Tulaczyk et al., 2014).

However, access technology using hot-water drilling systems has several significant drawbacks. For instance, these systems necessitate complicated methods to filter and subject high-speed hot-water flow (>150-200 L/min) to ultraviolet (UV) treatment at the surface. Additionally, they are extremely bulky and highly power-consuming. To simplify the drilling process and decontamination of drilling tools, we propose accessing and studying subglacial lakes with a freezing-in electric hot-point thermal drill — the RECoverable Autonomous Sonde (RECAS) — capable of downward and upward ice drilling and subglacial water sampling while ensuring that





the subglacial lake remains isolated from the surface (Talalay et al., 2014). RECAS is estimated
to be 10-20 times less expensive than penetration with a hot-water drilling system, and its
installation and operation require only four specialist staff members (Sun et al., 2023). The sonde
surface is thoroughly cleaned before deployment. Although the sonde might drag native microbes,
which are embedded in ice, into subglacial targets at various depths as they melt, this occurs in a
predictable manner (Schuler et al., 2018). Two concepts similar to RECAS have been proposed by
Stone Aerospace, a US engineering company (Pereira et al., 2023; Stone et al., 2018), and Aachen
University in Germany (Heinen et al., 2021).
The RECAS was successfully tested in East Antarctica during the 2021-2022 field season,
reaching the ice-sheet base at a depth of 200.3 m, sampling basal meltwater and measuring its
pressure, temperature, pH, and conductivity before returning to the ice surface (Sun et al., 2023).
To expand the sonde's possibilities, we propose equipping it with a steering technique to control
and guide the drilling process. This allows drilling at specific angles, depths, and directions,
enabling the sonde to follow a desired trajectory (e.g., maintaining verticality within the desired
range) or bypass obstacles in the ice (e.g., stones and other inclusions). Herein, we describe the
general principles of the steering RECAS and discuss the experimental results on deviational ice
drilling with a controllable electric thermal head.
**2. Steering approaches of the RECAS**
*2.1. General concept of the steering RECAS*
The RECAS comprises four major systems: a heating system (consisting of an upper melting
head, a lower melting head, and lateral heaters), an inner winch system, a scientific load platform,
and a parameter detection and control system (Sun et al., 2024). The upper and lower thermal
heads are identical except for the central hole of the cable in the top thermal head (Li et al., 2020).
Thus, it can drill both downward and upward and move within the borehole using an inner cable-
recoiling mechanism, similar to how a spider climbs on its silk line.





Two RECAS prototypes were developed: RECAS-200, with a 200-m-long cable inside, and
RECAS-500, with a 500-m-long cable inside. The RECAS-500 design is shown in Fig. 1. The
prototypes differed not only in their drilling ability but also in their sizes, power consumption,
number of cartridges in the thermal drill head, etc. (Table 1). In both prototypes, all heaters are
supplied simultaneously at the same voltage from a single source.
**Table 1**
General parameters of RECAS-200 and RECAS-500

| Prototype | Diameter, mm | Total length, m | Total power, kW | Power of thermal head, kW | Num. of cartridges in thermal head |
|-----------|--------------|-----------------|-----------------|---------------------------|-------------------------------------|
| RECAS-200 | 160 | 7.9 | 8.8 | 5 nom.; 6 max. | 16 |
| RECAS-500 | 180 | 7.3 | 9.7 | 6.5 nom.; 9.5 max. | 20 |

The fundamental principle behind an adjustable electric thermal head is to control the voltage
supplied to each pair of adjacent heaters. This enables control over the heat distribution on the drill
head surface and allows borehole deviations during drilling. Furthermore, controlling the heat
distribution of the thermal head makes it possible to equalise the load on the heaters as needed.
Heating cartridges exhibit variations in their parameters owing to their technological tolerances.
Additionally, these parameters may change slightly during long-term use, and heating cartridges
can fail because of their long-term use or manufacturing defects.
The sonde is steered using data from an inclinometer installed inside it. The data from the
inclinometer are transmitted to a personal computer (PC), processed, and converted into pulse-
width modulation (PWM) coefficients, which determine the PWM duty cycle for a specified
number of channels. In the subsequent tests using the RECAS-200 prototype, the PC will be
replaced with a microcontroller mounted inside the sonde. The PMW coefficients are transmitted
from the computer to a PWM generator (Sup. 2) inside the sonde prototype, where an individual
PWM signal is generated for each channel. Each PWM signal is amplified using a power module
(Sup. 3) and supplied to the corresponding heater inside the drill head. The PWM signal duty cycle
limits the heater power.



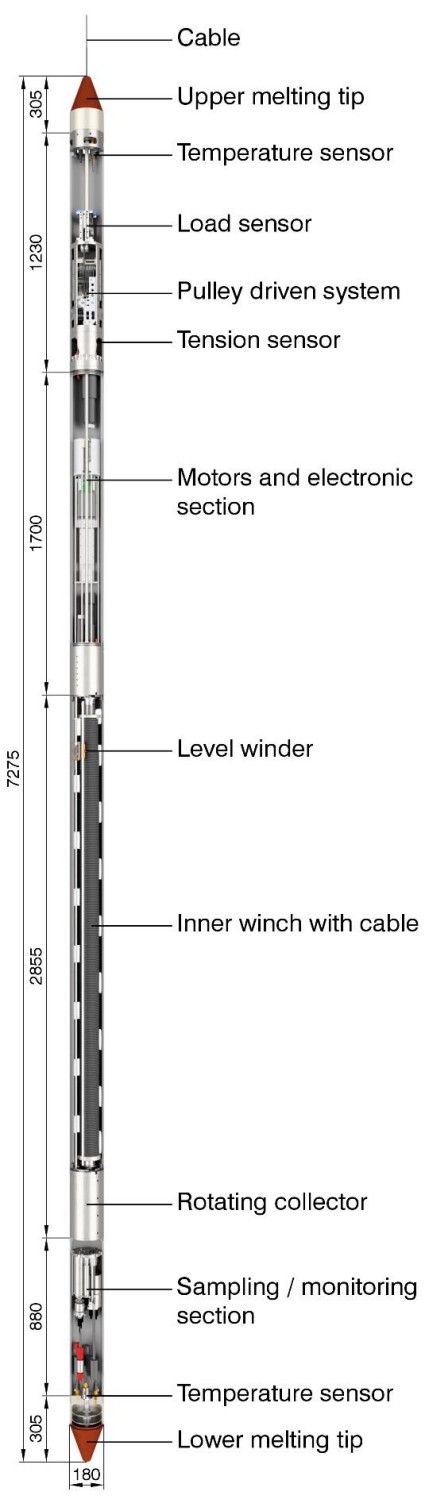


**Fig. 1.** General schematic of the RECAS-500 with a 500-m-long cable inside (all dimensions are in mm)



*2.2. RECAS positioning estimation*
The following method was employed to convert the values received from the inclinometer into
PWM coefficients. The inclinometer transmitted deviation values along the *X* and *Y* axes. As shown
in Fig. 2, the *X* and *Y* axes correspond to the inclinometer axes in the horizontal plane. Point *A*
indicates a deviation, with $X = 4.5$ and $Y = -3$, for instance. Eight pairs of 16 heaters are shown
schematically in the form of circles, designated as *H1-H8*.

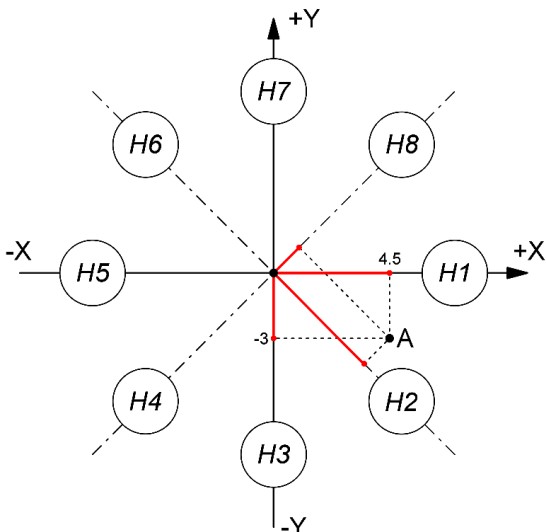


**Fig. 2.** Schematic diagram of the heaters *H1-H8* relative location and inclinometer in the sonde prototype.
First, the absolute inclination $\varphi$ is determined (Eq. 1). This value is required not only for
subsequent calculations but also for monitoring purposes.

$$\varphi = \sqrt{X^2 + Y^2} \tag{1}$$

where *X* and *Y* are the coordinates received from the inclinometer.
Next, the projection length values $l_n$ of point *A* on the axis of each pair of heaters are
determined as follows (indicated in red in Fig. 2):

$$l_n = \varphi \cos\left(\arctan\left(\frac{Y}{X}\right) - \alpha_n\right) \tag{2}$$

where $\alpha_n$ is the heater axis angle relative to the *X* and *Y* axes.





To obtain the required PWM coefficients, the $l_n$ values must be converted to relative values
in the range of 0-1. Additionally, it is necessary to be able to adjust the resulting coefficients. For
this purpose, a logistic function (logistic curve) was used (Kyurkchiev et al., 2015). After slight
adaptation to meet our conditions, the final equations take the following form:

$$K_n = \frac{1}{1 + \exp\left(-T\left(l_n + V\right)\right)}; \quad V = \frac{1}{T}\ln\frac{-y_{off}}{y_{off} - 1} \tag{3}$$

where $K_n$ is the PWM coefficient for each heater pair, $V$ is the intermediate coefficient, $T$ is the
correction coefficient (above zero), and $y_{off}$ is the offset coefficient (0-1).
$T$ and $y_{off}$ are used to adjust the final values. The coefficient $y_{off}$ limits the maximum average
PWM coefficient value (i.e., with zero inclination and $y_{off} = 0.5$, all PWM coefficients will be 0.5).
Meanwhile, the correction coefficient $T$ affects the rising section length where the derivative is
relatively large. The influence of $T$ and $y_{off}$ on the final results is illustrated by the example
discussed next.
*2.3. RECAS positioning calculation example*
For the calculation example, random inclinometer values are taken as: $X = 4.5$; $Y = -3$. Then,
absolute inclination is
$\varphi = \sqrt{X^2 + Y^2} = \sqrt{4.5^2 + \left(-3\right)^2} = \sqrt{20.25 + 9} = \sqrt{29.25} = 5.41$.
The $\alpha$ values for eight heater pairs are presented in Table 2.
**Table 2**
The $\alpha$ values for eight pairs of heaters

| $\alpha_1$ | $\alpha_2$ | $\alpha_3$ | $\alpha_4$ | $\alpha_5$ | $\alpha_6$ | $\alpha_7$ | $\alpha_8$ |
|---|---|---|---|---|---|---|---|
| 0° | 45° | 90° | 135° | 180° | 225° | 270° | 315° |

The projection length of the first heater is estimated as
$l_1 = \varphi\cos\left(\arctan\left(\frac{Y}{X}\right) - \alpha_1\right) = 5.41 \cdot \cos\left(\arctan\left(\frac{-3}{4.5}\right) - 0\right) = 5.41 \cdot \cos\left(-33.69\right) = 5.41 \cdot 0.83 = 4.5$.
The calculation results for all eight projection length values are listed in Table 3 and shown
in graph form in Fig. 3.
**Table 3**
Projection length values

| $l_1$ | $l_2$ | $l_3$ | $l_4$ | $l_5$ | $l_6$ | $l_7$ | $l_8$ |
|-------|-------|-------|-------|-------|-------|-------|-------|
| 4.5 | 1.06 | -3 | -5.3 | -4.5 | -1.06 | 3 | 5.3 |

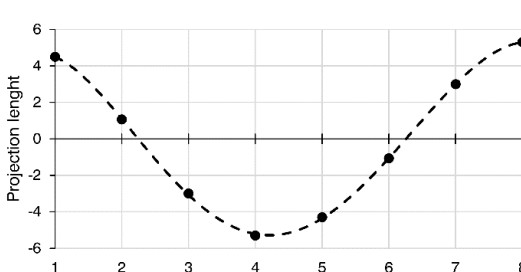

**Fig. 3.** Projection length values.
For this example, the following coefficients were selected: $T = 1$ and $y_{off} = 0.8$. Then, the
intermediate and PWM coefficients for the first heater pair are
$$V = \frac{1}{T} \ln \frac{-y_{off}}{y_{off} - 1} = \frac{1}{1} \cdot \ln \frac{-0.8}{0.8 - 1} = \ln 4 = 1.39 \; ;$$
$$K_1 = \frac{1}{1 + \exp(-T(l_1 + V))} = \frac{1}{1 + \exp(-1 \cdot (4.5 + 1.39))} = \frac{1}{1 + \exp(-5.89)} = \frac{1}{1 + 0.0028} = 0.997 \; .$$
The results of the final PWM coefficient calculations are listed in Table 4.
**Table 4**
PWM coefficients

| $K1$ | $K2$ | $K3$ | $K4$ | $K5$ | $K6$ | $K7$ | $K8$ |
|------|------|------|------|------|------|------|------|
| 0,997 | 0,92 | 0,17 | 0,02 | 0,04 | 0,58 | 0,99 | 0,999 |

To illustrate how $T$ and $y_{off}$ affect the final PWM coefficients, the calculation results with the
same deviation values but different $T$ and $y_{off}$ values are presented in Figs. 4 and 5.

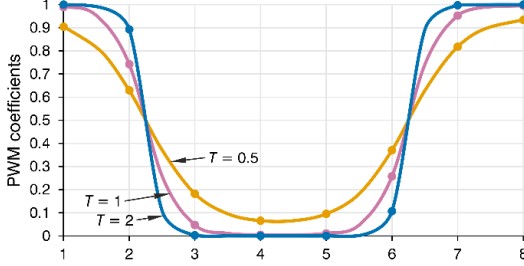

**Fig. 4.** PWM coefficients at constant $X = 4.5$; $Y = -3$; $y_{off} = 0.5$, and three different values of $T = 0.5$; $T = 1$; $T = 2$.

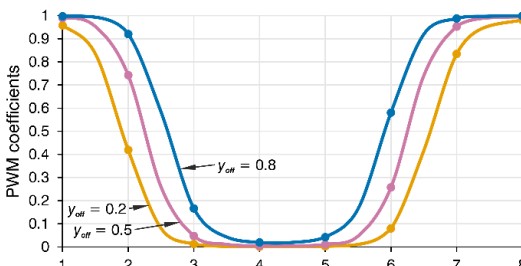


**Fig. 5.** PWM coefficients at constant $X = 4.5$; $Y = -3$; $T = 1$, and three different values of $y_{off} = 0.2$; $y_{off} = 0.5$; $y_{off} = 0.8$.

The coefficients $T$ and $y_{off}$ should be selected experimentally. Therefore, they do not need to
remain constant. They may depend on other parameters; for example, $y_{off}$ may depend on the
absolute inclination. It is worth noting that when the correction coefficient $T$ approaches zero, all
PWM coefficients tend towards the $y_{off}$ value, which means that the heat distribution on the drill
head surface approaches a uniform pattern.
**3. Passability of the RECAS**
Before changing the borehole trajectory direction, determining the passability of the sonde
in the drilled borehole is essential. Owing to its length exceeding 7 m, the RECAS has a high
likelihood of becoming stuck in the borehole, even with relatively small deviations. The main
parameter affecting sonde passability in a curved borehole is the deviation intensity. To
characterise the borehole deviation intensity at a specific interval along its axis, the relative zenith
angle values were used, considering the interval between their measurement points. Therefore, the
zenith deviation intensity was determined as follows (Zvarygin, 2010; Shamshev et al., 1983):

$$i_\theta = \frac{\Delta\theta}{\Delta L} \tag{4}$$

where $\Delta\theta$ is the relative zenith angle in degrees and $\Delta L$ is the borehole axis interval length.
If the deviation intensity at some borehole interval remains constant ($i_\theta = $ const), it means
that the borehole is curved along a circular arc at a certain interval. The borehole radius of
curvature $R$ depends on the deviation intensity, as follows (Fig. 6):

$$R = \frac{57.3}{i_\theta} \tag{5}$$



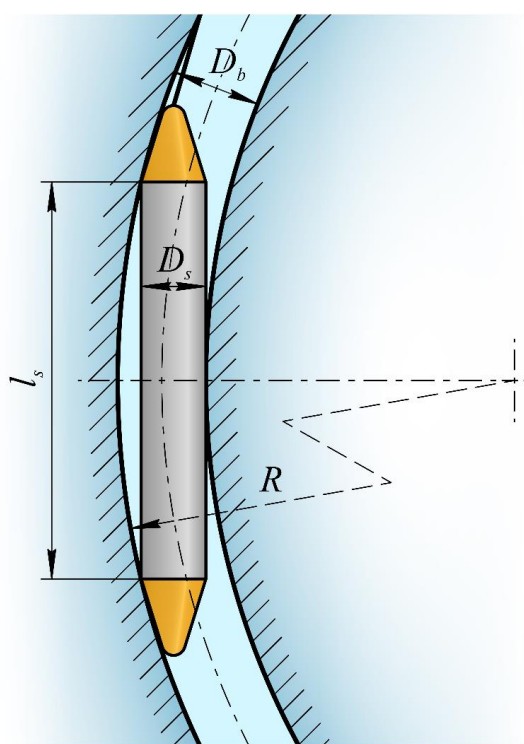


**Fig. 6.** Schematic of the stuck sonde in a curved borehole.
The passability of a sonde in a borehole interval with known diameter and radius of curvature
can be determined as follows:

$$l_s \leq \sqrt{8R\left(D_b - D_s\right) - 4\left(D_s - D_b\right)^2} \qquad (6)$$

where $l_s$ is the length of the cylindrical part of the sonde (the thermal head length is not included),
$D_s$ is the sonde diameter, and $D_b$ is the borehole diameter.
As the borehole radius of curvature is considerably larger than the gap between the sonde
and borehole diameters, Eq. 6 can be simplified as follows (Shamshev et al., 1983):

$$l_s \leq \sqrt{8R\left(D_b - D_s\right)} \qquad (7)$$

Based on RECAS field tests, the difference between the borehole and sonde diameters are
10-20 mm. This clearance mainly depends on the rate of penetration (ROP), and additional
laboratory tests are required to establish a more precise relationship. Considering that the RECAS





length is approximately 7 m, the range of radii of curvature ensuring RECAS passibility is in the
range of 300-600 m.

Therefore, it is not sufficient to simply monitor the borehole inclination to avoid the RECAS

from being stuck in the borehole. Instead, it is necessary to continuously estimate the deviation
intensity, the borehole radius of curvature, or both at an interval from the bottom hole with a length
approximately equal to that of the sonde.
**4. Testing stand and sonde prototype design**
*4.1. Testing stand*
*4.1.1. General testing stand design*

The testing stand consists of a sledge, mast, top wheel, winch, and sonde prototype (Fig. 7).

All stand parts are mounted on a sledge, which has a modular construction comprising a pair of
skis and two welded frames bolted together. A 2-m-high mast is mounted in the middle of the
sledge. A small winch is mounted near the mast on a sledge. A block is installed at the top of the
mast. The testing stand parameters are listed in Table 5.
**Table 5**
Testing stand parameters

| | |
|---|---|
| Mast height | 2 m |
| Max length of the testing sonde | ~2 m |
| Weight of the testing sonde | nom. 100 daN or less; max. 200 daN. |
| Max. volume of the winch drum | 10 m length of 5 mm Kevlar cable |
| Min. ROP | 0.1 m/h (*ROP values refresh rate no more than once per ~6.5 sec*)<br>0.72 m/h (*ROP values 1 sec. refresh rate*) |
| Max. possible tripping speed | 9.3 m/min (*weight of the testing sonde no more than 57 daN*) |
| Max. tripping speed | 5 m/min (*for 100 daN testing sonde*)<br>2.3 m/min (*for 200 daN testing sonde*) |





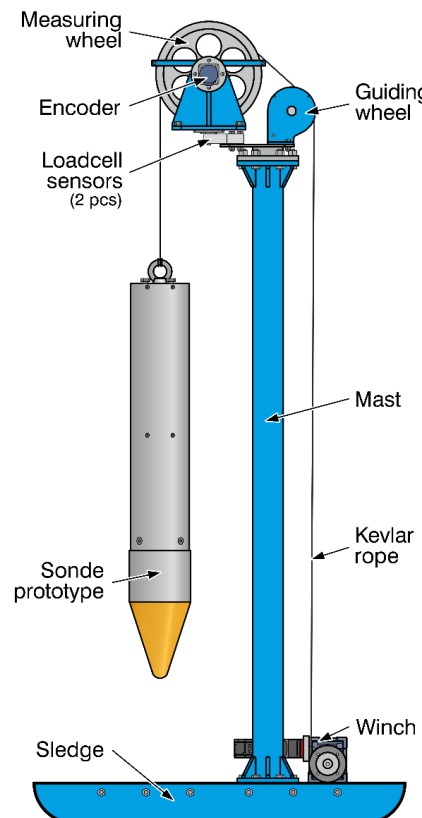

**Fig. 7.** Schematic of the testing stand

*4.1.2. Top block and as-low-as-practicable ROP*
The top block consists of two wheels – a measuring wheel and a guiding wheel, an encoder,
and two load-cell sensors. The measuring wheel is designed for a rope with a 5 mm diameter so
that the cable length passing through the wheel per revolution equals 1 m ±1 mm. This design
simplifies the calculation and further adjustment of the measuring equipment. The guiding wheel
is used to guide the rope from the winch to the measuring wheel.
To register the weight on bit (WOB), two load-cell sensors are installed underneath the top
wheel. Each sensor can withstand a force of up to 100 daN. To measure the ROP, an encoder with
an accuracy of 5000 measurements per revolution (MPR) is installed on the measuring wheel shaft.
As the ROP is expected to be relatively low, the angular rotation speed of the measuring wheel is



correspondingly small. Therefore, the higher the encoder accuracy, the more frequently it can
capture instantaneous low-ROP values.

As-low-as-practicable ROP [m/h] can be estimated as follows:

$$\upsilon_{min} = 3600 \cdot \pi (D + d) n_{min} \qquad (8)$$

where $D$ is the wheel diameter ($D = 0.3135$ m); $d$ is the rope diameter, and $n_{min}$ is the minimum angular
velocity in revolutions per second (RPS)

Minimal angular velocity $n_{min}$ is equal to:

$$n_{min} = \frac{1}{tm} \qquad (9)$$

where $m$ is the encoder accuracy in MPR and $t$ is the time after which the data must be updated (in
this study, $t = 1$ s).

The wheel diameter is:

$$D = \frac{l}{\pi} \qquad (10)$$

where $l$ is the wheel circumference.

After all rearrangements, the as-low-as-practicable ROP is:

$$\upsilon_{min} = 3600 \cdot \frac{l}{tm} \qquad (11)$$

Therefore, the as-low-as-practicable ROP, which can be measured with a 5000-MPR encoder,

1-m wheel circumference, and a measurement frequency of once per second, was 0.72 m/h.
*4.1.3. Winch*

The winch is based on an RV50 series worm gearbox. For precise winch control, a 200-W

power servo drive was chosen in this study. To compensate for the low servo power, a small
PX60 series planetary gearbox with a gear ratio of 1:6 was installed between the worm gearbox and
servo. The small drum was customised to hold one layer of 5-mm-diameter Kevlar rope with a length
of 10 m. To simplify the winch construction, the drum was mounted directly on the output shaft of
the worm reducer. Further details regarding the winch construction design are presented in Sup. 1.





### 4.1.4. Control system

The control system consists of a box containing various data acquisition modules (Fig. 8). Data acquisition modules ADAM 4017+ and ADAM 4018+ were used to collect data from the load-cell sensors and thermocouples, respectively. Counter-tachometer module CTA4001A was used to receive and convert signals from the encoder on the measuring wheel. Two MIK-1100 modules were connected to voltage and current sensors. Temperature module DT320 was used to monitor the sonde prototype drill head temperature. All modules, along with the voltage and current sensors, were mounted in a BDH20016 black case. The wiring schematics for all components are shown in Fig. S5.1 (Sup. 5). The sensor parameters are listed in Table 6.

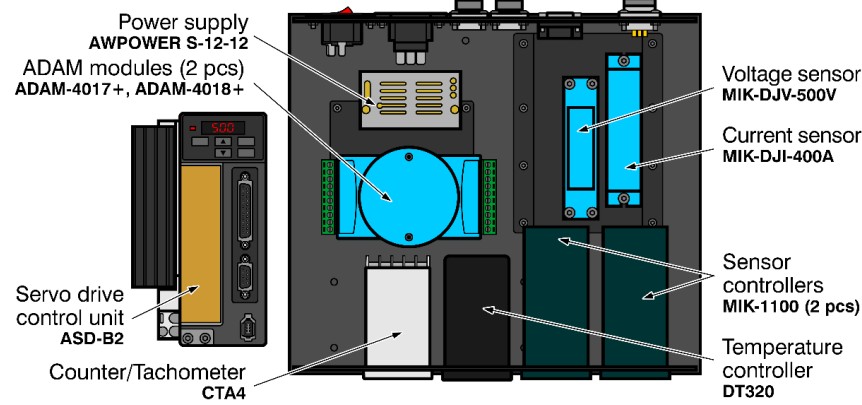

**Fig. 8.** Schematic of control system

**Table 6**

Parameters of the sensors

| Sensor type | Range | Accuracy | Mounting location | Meas. values |
|---|---|---|---|---|
| Encoder BC58S10 | up to 6000 RPM | 5000 MPR | Top block | ROP, Depth |
| Load-cell YZC-320C (2 pcs) | up to 100 kg | ≤±0.02% | Top block | WOB |
| Voltage sensor MIK-DJV-500V | up to 500V | 0.2% | Control system box | Voltage |
| Current sensor MIK-DJI-400A | up to 400A | 1% | Control system box | Current |
| T type thermocouples | from -270°C up to 370°C | ±0.75% | Ice block, air, drill head and control system box | Temperature |





### 4.1.5. Software

The control system box, servo control unit, and drill head control unit were connected to a computer via RS-485. The MODBUS RTU communication protocol was used for data transmission. The software registers the following parameters from the sensors connected to the control system box: ROP (m/h) and Depth (m) (Section 2 in Fig. 9); WOB (daN) (Section 3); Current (A), Voltage (V), Power (W), and three temperatures (°C) (Section 5). Through the drill head control unit, the software allows monitoring of the sonde prototype inclination and heater status and allows selection between manual and automatic modes (Section 4). A control panel for the winch is located at the bottom of the screen (Section 6).

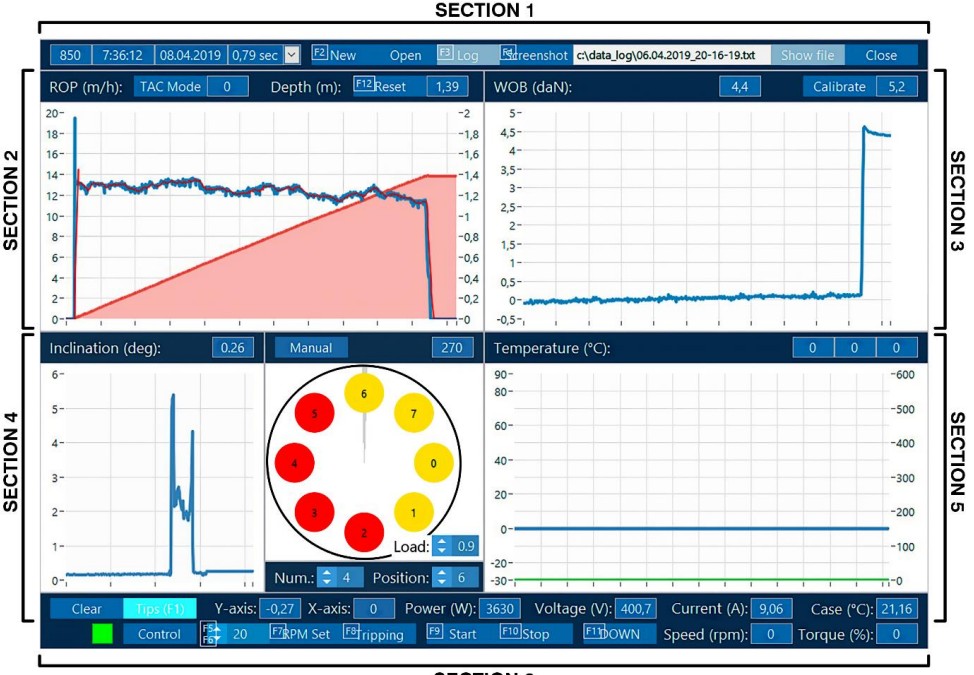

**Fig. 9.** Software main screen

*4.2. Testing sonde prototype*

*4.2.1. General structure of the sonde prototype*

The sonde prototype consisted of a thermal drill head borrowed from the RECAS-200 prototype and a control unit assembled inside the housing (Fig. 10). The total sonde prototype



length was approximately 1.1 m, and its weight was approximately 35 kg. The sonde prototype
was suspended using a Kevlar rope tied to a hook. Electric lines for the power supply and
communication were inserted through isolated connectors in the top cover.

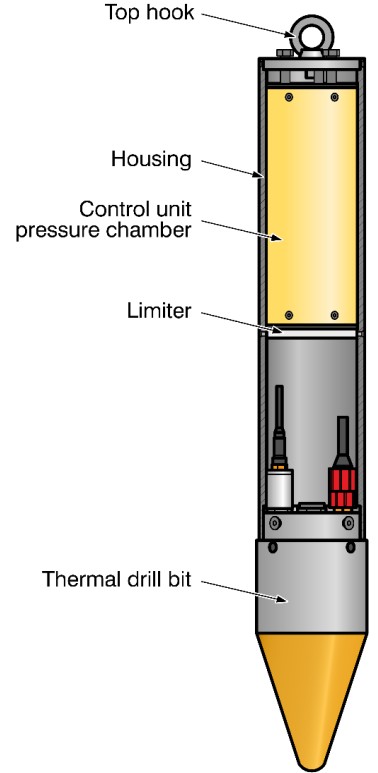

**Fig. 10.** Schematic of the self-steering sonde prototype

*4.2.2. Thermal drill head*
The thermal head diameter was 160 mm. It had 16 heat cartridges with a total power of
approximately 7.6 kW (Li et al., 2020; Talalay et al., 2019). The heaters connections in the
thermal drill head were redesigned (Fig. 11). Fuses were installed on each heater, and a distribution
board was designed to distribute the load and connect it to the power connector (Fig. 12). To allow
each heater to be individually connected to a power source, the power connector was also changed
from two four-pin connectors to one 21-pin connector.


(a) (b)

**Fig. 11.** Thermal drill head: (a) schematic and (b) photo

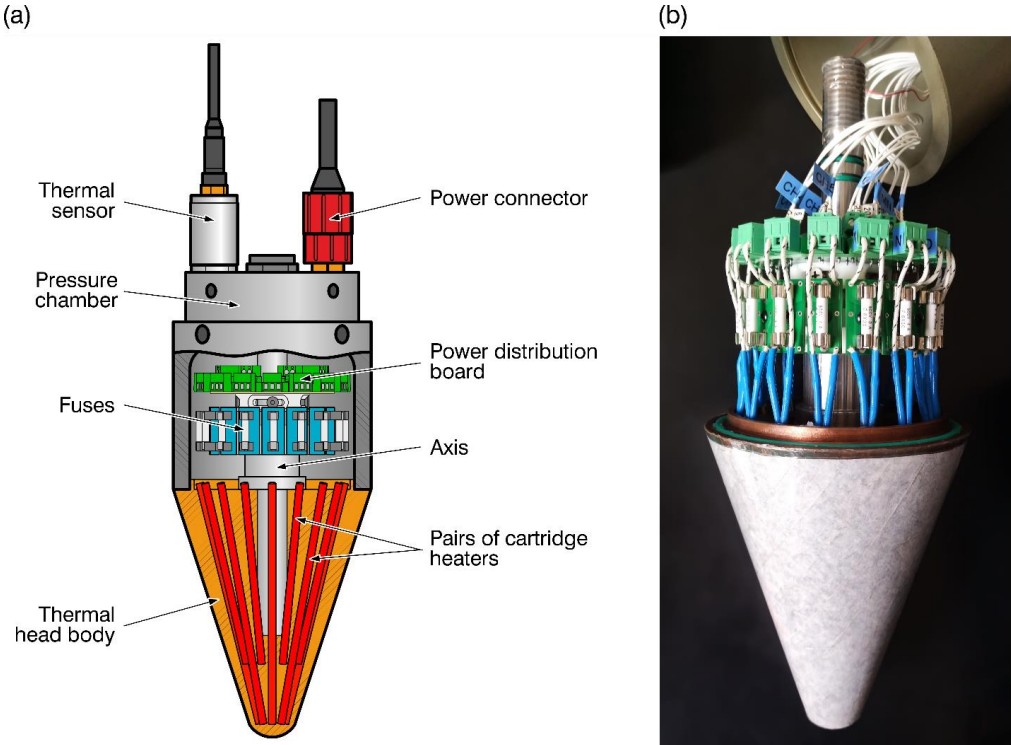

**Fig. 12.** Electrical schematic of thermal drill head

The thermal drill head uses eight long heaters (200 mm in length) and eight short heaters (150 mm in length) that were arranged in an alternating pattern. The cartridges were controlled in pairs, with each long heater paired with an adjacent short one (Fig. 12). Therefore, the number of required PWM signals (PWM channels) was reduced to eight.





*4.2.3. Control unit*
To control the heaters in the sonde prototype, a control unit was designed as a pressure
chamber housing the following components (Fig. 13).

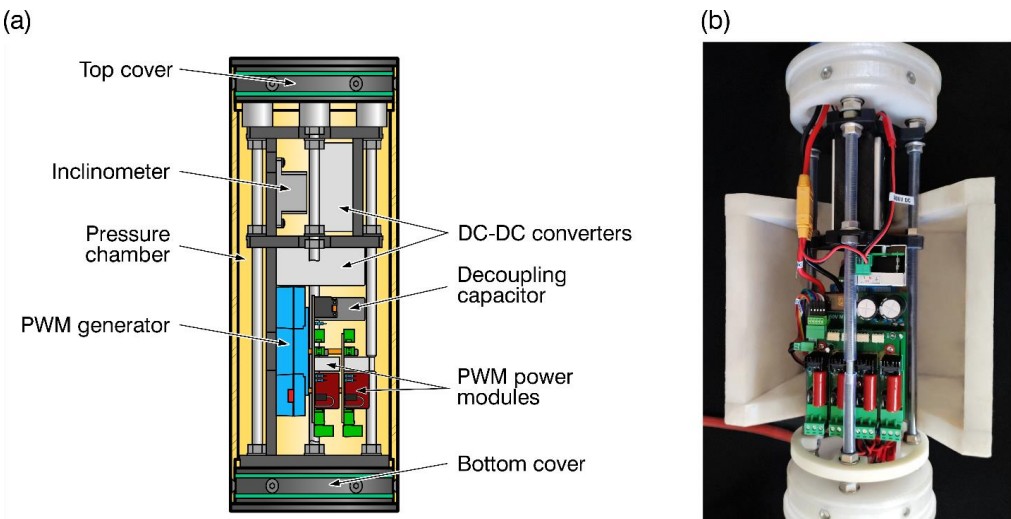

**Fig. 13.** Control unit: (a) schematic and (b) photo
*Dual axis inclinometer*. A two-axis inclinometer was chosen instead of a three-axis one
because the control unit was rigidly fixed together with the thermal drill head inside the sonde
prototype, eliminating the need to track the relative rotation along the vertical axis. The task of
tracking the borehole azimuth is planned for future RECAS prototype testing.
*PWM generator for 20 channels* (Sup. 2). Although only eight PWM channels were required
in this study, the PWM generator was designed with 20 channels to enable the control of individual
heaters in the bottom drill head in the RECAS prototype in the future.
*Two power modules, 4 channels each* (Sup. 3). The heater pairs were not connected directly
to the PWM generator but through power modules that amplify the corresponding PWM signals
from the generator.
*Two 15-volt DC-DC converters* (Sup. 4). Two identical DC-DC converters were used to
isolate the power supply of the inclinometer and the PWM generator from the power supply of the
low-voltage part of the power modules.
The wiring schematics for all components are shown in Fig. S5.2 (Sup. 5).





All control unit modules, except for the inclinometer, were customised for this study. The
primary characteristics of the control units are listed in Table 7.
**Table 7**
Control unit main parameters

| Parameter | Value |
|---|---|
| Power supply | 100-500 V DC |
| Limit values for angle measurement | $X$ axis $\pm90$ $Y$ axis $\pm45°$ |
| Angle measuring accuracy | 0.2° |
| Number of PWM channels | 8 (upgradeable to 20 Ch.) |
| Communication with PC | RS-485 MODBUS RTU |

Further details regarding each individually designed module can be found in the
corresponding supplements.
**5. Laboratory testing of self-steering sonde prototype**
*5.1. Factors and main parameters of experiments*
To study the sonde prototype inclination in a drilled borehole, a series of tests were
conducted in the Polar Research Center laboratory.
The main factors affecting the sonde inclination and drilling performance are:
1. Ice temperature (kept constant at −16°C);
2. Environmental temperature (varied slightly between +7°C and +12°C);
3. ROP, which was controlled by the winch and limited by the power supplied to the heaters
inside the drill head;
4. WOB, which changed with the ROP and was limited by the sonde prototype weight.
The main parameter to be recorded was the sonde inclination. The sonde inclination was
affected by the controlled heat distribution on the drill head surface, which was controlled by
limiting the heater power. The control algorithm, with two variable coefficients $T$ and $y_{off}$ is
described in Eq. 3. Therefore, the main purpose of the experiments was to determine the
dependence of the sonde inclination on these coefficients. For clarity and visual control, blocks of
transparent ice with dimensions of $50 \times 50 \times 100$ cm were used in the experiments (Fig. 14).





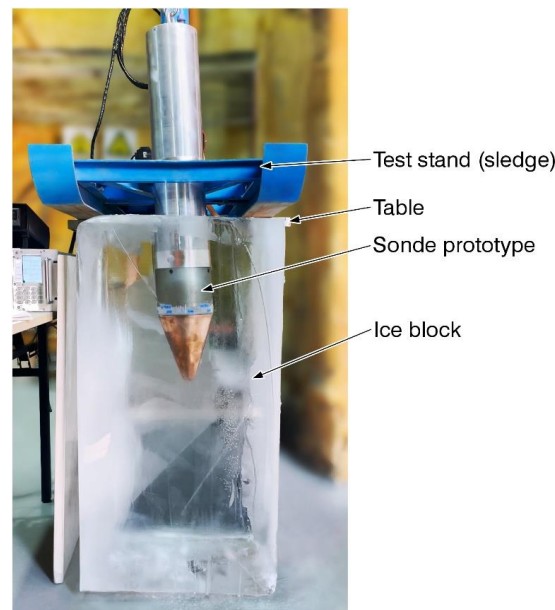

**Fig. 14.** Testing of the sonde prototype with RECAS-200 controllable thermal head

*5.2. Preliminary experiment*

A preliminary experiment was conducted to determine the potential ROP and WOB ranges

and the possibility of controlling the sonde prototype inclination in the borehole by regulating the
heater power in the drill head. The experiment recording was divided into four sections (Fig. 15).

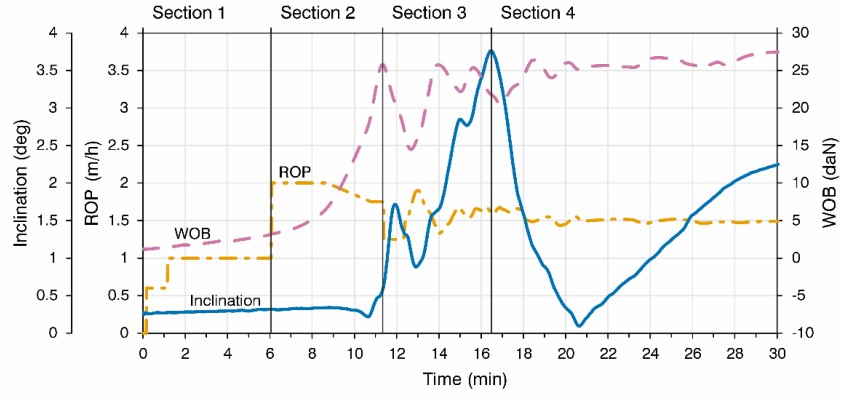


**Fig. 15.** Recording of the preliminary experiment





*Section 1 (0-6 min).* The first 200 mm of drilling were strictly vertical, with an ROP of
0.6 m/h and all heaters running at 50% power. Subsequently, the power was increased, and the
ROP was increased to 1 m/h. Then, half of the heaters on one side of the thermal head were
switched off.
*Section 2 (6-11 min).* No sonde prototype inclination was detected, and the ROP was set to
2 m/h to increase the WOB.
*Section 3 (12-16 min).* The inclination began to increase rapidly when the WOB value
reached approximately 25 daN. An attempt was made to stabilise the WOB at this value. The WOB
stabilised at approximately 25 daN with an ROP of approximately 1.5 m/h.
*Section 4 (16-30 min).* When the sonde prototype inclination angle reached
approximately 4°, the powered heaters configuration was changed. Four previously powered
heater pairs were switched off, and four heater pairs on the opposite side were switched on.
Based on the preliminary experimental results, the following conclusions can be drawn. To
achieve the desired sonde prototype inclination, the WOB should be approximately 20 daN or
higher. However, the test sonde weight was only 35 daN, which significantly reduced the range of
the acceptable WOB values. To avoid a situation in which the entire sonde prototype weight would
be at the bottom of the borehole, the WOB range was limited to 22-28 daN.
When a 50% power limit was applied, the WOB stabilised at an ROP of approximately
1.5 m/h. Although WOB is not directly controlled, it depends on the ROP. However, constant
WOB adjustments via ROP changes using a proportional-integral-derivative (PID) controller were
not very effective because the transients significantly influenced the measured parameters.
Therefore, in subsequent experiments, we decided to maintain a constant ROP despite potential
WOB fluctuations.
Because the coefficient $y_{off}$ limits the maximum average PWM coefficient values, in practice,
it limits the power consumption of the drill head, which, in turn, affects the maximum ROP.
Preliminary experimental results showed that testing was meaningful only at WOB values close



359 to the maximum. This means that at a certain $y_{off}$ value, it is not possible to change the ROP over

360 a significant range. By analysing the "behaviour" of Eq. 3 we can conclude that at $y_{off} = 0.5$, the

361 heat distribution is the most intense, and the assumed rate of change in borehole trajectory is also

362 at its maximum. The limitation of the maximum borehole depth that can be obtained from the

363 available ice blocks underscores its importance.

364  Based on the above, we conducted a series of experiments with four different correction

365 coefficient *T* values. The ROP was kept constant at 1.5 m/h. The WOB stabilised between 22 and

366 28 daN. The power consumption was limited to 50% by setting $y_{off} = 0.5$.

367  According to the test plan, in the first approximately 300 mm of each experiment, the sonde

368 prototype should drill with half of the heaters on one side turned off until the sonde inclination

369 angle reaches approximately 4° (Ye et al., 2024). Subsequently, the automatic alignment mode

370 will be enabled. The algorithm will recalculate the PWM coefficients of the heaters at 1-s intervals.

371 It is worth noting that decreasing the PWM coefficient recalculation frequency (i.e., slowing the

372 response to inclination angle changes) can influence the borehole deviation intensity. A decrease

373 in the recalculation frequency is likely to result in a decrease in borehole deviation intensity.

374 *5.3. Experimental results and analysis*

375  A total of four experiments were performed. The experimental recordings are shown in

376 Fig. 16. As the values were recorded from the sensors at a frequency of once per second, the graphs

377 are depicted with a 15-value moving-average filter. The graphs show trend lines for each

378 experiment. For each trend line, the bold line indicates the accuracy limits according to the

379 inclinometer specifications (±0.2°).

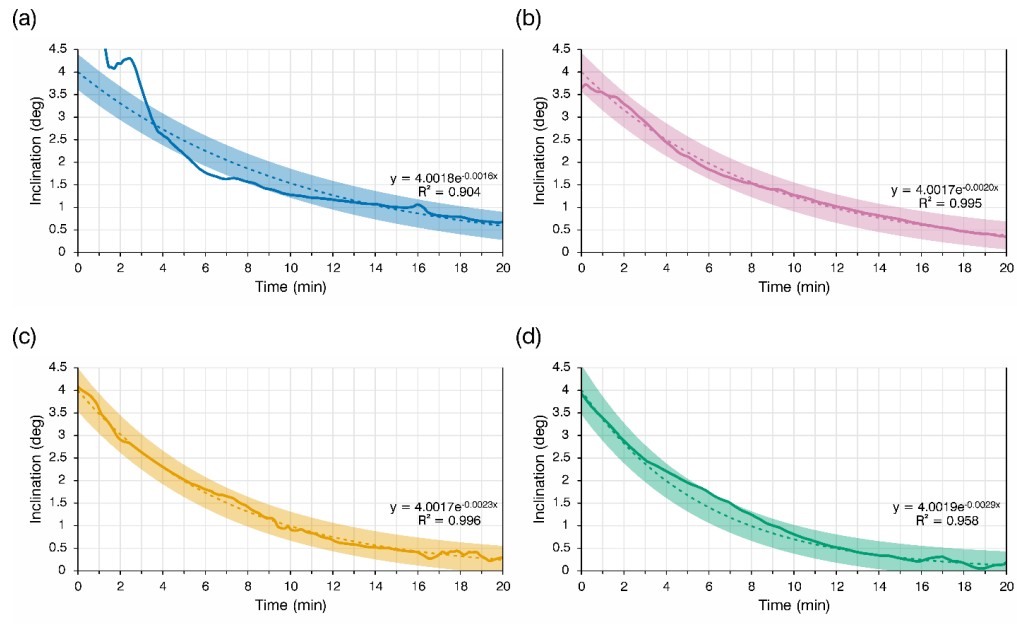

**Fig. 16.** Recording of the four experiments with trendlines:
(a) Borehole 1, $T = 0.5$, (b) Borehole 2, $T = 1$, (c) Borehole 3, $T = 1.5$ and (d) Borehole 4, $T = 2$

Drilling of borehole 1 with $T = 0.5$ was unsuccessful owing to water leakage from the borehole; consequently, the results were difficult to analyse. The graph illustrates an approximation option intended to be obtained based on the analysis of the other three experiments. The experiment demonstrates that correction coefficient $T$ affected how rapidly the borehole deviation changed over time. For clarity, the approximations of all four experiments are shown in Fig. 17.

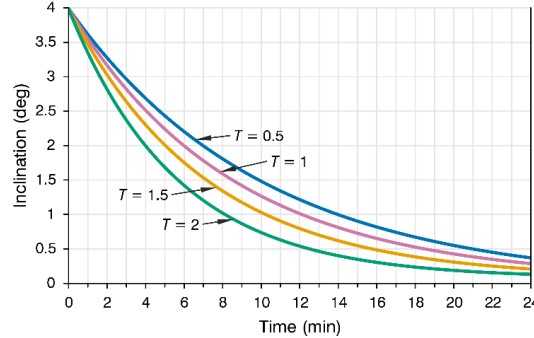

**Fig. 17.** Approximations of four experiments

To understand how crucial borehole deviations are for the passability of the sonde prototype, it is necessary to analyse the results for the allowable deviation intensity according to the method





described in Section 3. In the experiments, the automatic alignment length was
approximately 0.5 m. For clarity, it was decided to divide this 0.5-m section of each borehole into
several sections, for each of which the radius of curvature was determined. The option of
partitioning the path not according to depth but rather according to inclination angle proved to be
the most illustrative. Four sections were selected with the following inclination angle ranges:
[4°-2.5°], [2.5°-1.5°], [1.5°-1°], and [1°-0.5°].

The radius of curvature was determined for all four boreholes in each of the selected sections.

To determine the radius, an additional angle (approximately in the middle of the range) was
selected. The following additional angle values were selected for further calculations: 3.25° for
the range [4°-2.5°], 2° for [2.5°-1.5°], 1.25° for [1.5°-1°], and 0.75° for [1°-0.5°]. Fig. 18 shows
the sonde trajectory for the [4°-2.5°] range in borehole 2 ($T = 1$).

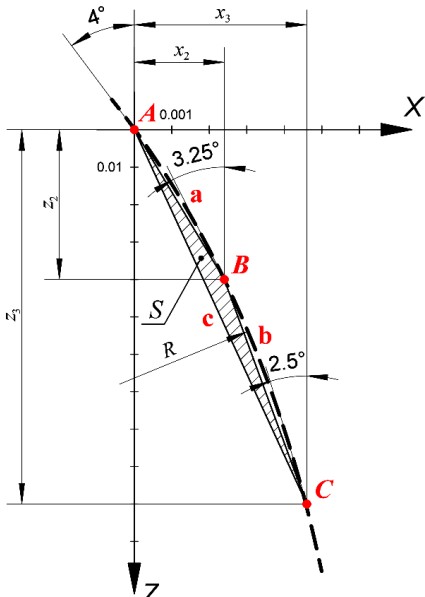


**Fig. 18.** Borehole 2 ($T = 1$) trajectory within inclination angle range of [4°-2.5°]

The sectional radius of curvature was calculated as follows:

$$R = \frac{abc}{4S} \tag{12}$$

where $a$, $b$ and $c$ are the side lengths of triangle $ABC$ and $S$ is the area of triangle $ABC$.





The area of the triangle can be determined using Heron's equation:

$$S = \sqrt{p(p-a)(p-b)(p-c)} \tag{13}$$

where $p$ is the semi-perimeter of a triangle.
The side lengths of triangle $ABC$ can be determined using the Pythagorean theorem given
the coordinates of the points $A$ $(x_1, z_1)$, $B$ $(x_2, z_2)$ and $C$ $(x_3, z_3)$ are known:

$$a = \sqrt{(x_2 - x_1)^2 + (z_2 - z_1)^2} \tag{14}$$

$$b = \sqrt{(x_3 - x_2)^2 + (z_3 - z_2)^2} \tag{15}$$

$$c = \sqrt{(x_3 - x_1)^2 + (z_3 - z_1)^2} \tag{16}$$

Knowing the coordinates of points $A$, $B$, and $C$ for all sections using Eqs. 12-16, the radius
of curvature of each section can be determined. Substituting the radius of curvature into Eq. 6, the
maximum allowable sonde length satisfying the passability criteria for each segment can be
determined. The resulting radius of curvature and maximum allowable sonde length values are
presented in Table 8 and Fig. 19.
**Table 8**
Radius of curvature and maximum allowable sonde length

| Range | Borehole 1, $T = 0.5$ | | Borehole 2, $T = 1$ | | Borehole 3, $T = 1.5$ | | Borehole 4, $T = 2$ | |
|---|---|---|---|---|---|---|---|---|
| | Radius of curvature (m) | Length of sonde (m) | Radius of curvature (m) | Length of sonde (m) | Radius of curvature (m) | Length of sonde (m) | Radius of curvature (m) | Length of sonde (m) |
| 4°-2.5° | 4.701 | 0.866 | 3.761 | 0.775 | 3.270 | 0.722 | 2.594 | 0.643 |
| 2.5°-1.5° | 7.64 | 1.105 | 6.112 | 0.988 | 5.315 | 0.921 | 4.215 | 0.82 |
| 1.5°-1° | 12.115 | 1.392 | 9.692 | 1.245 | 8.428 | 1.161 | 6.684 | 1.033 |
| 1°-0.5° | 20.7 | 1.819 | 16.56 | 1.627 | 14.4 | 1.517 | 11.421 | 1.351 |

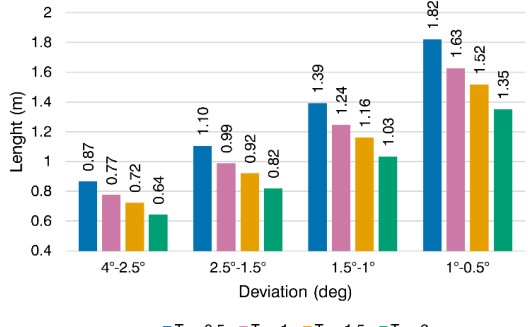

**Fig. 19.** Dependence of the maximum allowable sonde length at different borehole sections with the same inclination change



It is worth noting that the difference between the initial and final angles has a significant
impact on the radius of curvature. If the analysed section tends to zero, the radius of curvature of
such a section tends to infinity, and vice versa. As a compromise, the ranges were selected to
minimise the difference between the size of the deviation angles ranges and the corresponding
borehole section lengths.
**6. Conclusions**
Based on the experimental results for the sonde prototype, the main conclusions of this study
can be summarised as follows:
1. The sonde prototype demonstrates a promising potential in controlling the borehole direction

and using the RECAS, it should be possible to control the borehole direction to a certain extent

using the proposed method.

2. The borehole deviation intensity during drilling can be corrected by controlling the correction

coefficient $T$.

3. It is worth noting that the radius of curvature of a real RECAS would be higher than that

obtained experimentally. Further research is required to obtain the RECAS parameters.

However, to prevent the RECAS from becoming stuck in its own borehole at the chosen

experimental drilling parameters, the borehole deviation intensity must be reduced.

4. At the maximum borehole diameter value obtained in the field for the RECAS-200 prototype,

the maximum theoretical borehole deviation value cannot exceed 0.67° at a sonde length of

approximately 7 m. However, this calculation did not consider the fact that a 7-m-long sonde

may exhibit some deformability (especially at the joints), despite its housing being made of

stainless steel. At this length, even a small deformation of a few millimetres could positively

affect the passability of the sonde in the borehole.

5. Sonde passability at large borehole deviation intensity values can be improved if the housing

is structurally divided into several parts capable of deviating from each other (hinged joints).



Allowing just a half-degree deviation of one part of the sonde from the other could increase

its passability.

In future work, we plan to conduct experiments on a larger scale (e.g., with a borehole depth

of approximately 10 m) to refine the results in a deviation intensity range closer to that obtained
with a real RECAS.
**Data availability**

All raw data can be provided by the corresponding authors upon request.

**Author contributions**

Conceptualization: TPG, SMA, FX; hardware and equipment design: SMA, GD; software

development SMA; resources and supplies: FX, GD, DZ, ZN; planning the experiment: SMA, TPG;
assistance in preparing for experiments: FX, ZN, GD, DZ; conducting experiments and performed
the measurements: SMA, FX; analysing the data: SMA, TPG; project administration: FX, ZN;
financial management: YY, WT; supervision TPG; writing the manuscript draft: SMA, TPG;
reviewing and editing the manuscript TPG; reviewing the manuscript: FX, GD.
**Competing interests**

The authors declare that they have no conflict of interest.

**Acknowledgements**

This research was supported by the National Key Research and Development Project of the

Ministry of Science and Technology of China (Grants No. 2023YFC2812602, 2021YFC2801401)
and the National Natural Science Foundation of China (Grant No. 41941005). We thank all teachers,
engineers and postgraduate students at the Polar Research Center of Jilin University for their hard
work in developing and testing thermal sonde and solving various problems. We also thank the
members of the FagearTechCorner discord server community for their help in development and
fruitful suggestions.



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
