# Peer review of "Steering recoverable autonomous sonde (RECAS) for accessing and studying"

_Geoscientific Instrumentation, Methods and Data Systems, 2024_

## Referee Comment (RC2)

**Comments**

The manuscript presents the design and preliminary test results of a steerable system of RECAS for accessing and studying subglacial lakes. The designed sonde prototype can be used for directional ice drilling with a controllable electric thermal head, which is meaningful to enabling melt probe to follow a desired trajectory or bypass obstacles in the ice. Generally, the manuscript is well-written and can be published after minor revision.

1) The manuscript mainly discussed a steerable system developed for RECAS, which is not a real steering recoverable autonomous sonde (RECAS), so the title is suggested to be changed. For example, "*A steerable system of RECoverable Autonomous Sonde (RECAS) for accessing and studying subglacial lakes*" or "*A prototype of steering RECoverable Autonomous Sonde (RECAS) for accessing and studying subglacial lakes: Design and test*".

2) Line 11 and Line 23: "Thermal sonde", "hot-point" and "thermal drill" have similar meaning, please use the same name throughout the manuscript.

3) Line 56, 57 and 58: The melt probe IceMole can also perform directional drilling in ice, can you provide more information about the IceMole's design in thermal head and its drilling performance. (References: *Curvilinear melting – A preliminary experimental and numerical study; IceMole: a maneuverable probe for clean in situ analysis and sampling of subsurface ice and subglacial aquatic ecosystems*)

4) In the manuscript, there are many names of the thermal head, such as "melting head", "thermal head", "drill head", "melting tip", "thermal drill head", "thermal drill bit", please use the same name throughout the manuscript.

5) Line 171 and 175: Is there any reference for the formula 5 and 6? How do you get the formulas?

6) Line 184: If the radii of curvature ensuring RECAS passibility is 300-600 m, is it means that the RECAS is very difficult to bypass obstacles in the ice unless the RECAS start to deviate before a long distance to the obstacles? If yes, please clarify.

7) Figure 10: What is the function of the limiter?

8) Figure 11: It would be better if the photo can show the full thermal head, such as pressure chamber, thermal sensor and power connector etc.

9) Line 288: "Dual axis inclinometer" or "two-axis inclinometer"? Please use the same name.

10) Line 311: Polar Research Center laboratory of Jilin University?

11) Line 334: How much is the 50% power and how much power is increased?

12) Figure 15: It is better to have depth data in the figure.

13) Line 342-344: After the four heater pairs on the opposite side were switched on, the inclination angle decreased to nearly zero and then gradually increased in opposite direction? If so, please clarify.

14) Line 364-373: The two paragraphs should be part of the section of 5.3, which shows how you perform the four experiments. In section 5.2, only the preliminary test procedure and test results should be included.

15) Line 369: Please present more information about "automatic alignment mode"? How do you control the power? How the power changed?

16) Line 434: What kind of RECAS parameters?

17) Line 444: The steering capability of melt probe can be used to maintaining verticality within the desired range or bypass obstacles in the ice. However, according to the research, it looks that a long melt probe is difficult to bypass obstacles in the ice because of its large radii of curvature. In RECAS situation, do you have other methods to bypass obstacles in the ice except for hinged joints?

Technical issue:

In the manuscript, the word "deviational" was used to describe non-vertical ice drilling process. However, Zagorodnov use the word "directional". Please check the exact expression of this term.

(*Zagorodnov V S, Kelley J J, Koci B R. Directional drilling. Memoirs of National Institute of Polar Research, 1994, Special issue 49:165-171.*)

---

## Author Response (AR1)

**Response to Referee #1**

We sincerely thank the referee for positive feedback and recommendation for publication. Your encouraging comments are greatly appreciated.

**Response to Referee #2**

We sincerely thank the referee for detailed review and valuable feedback.

**1.) The manuscript mainly discussed a steerable system developed for RECAS, which is not a real steering recoverable autonomous sonde (RECAS), so the title is suggested to be changed. For example, "A steerable system of RECoverable Autonomous Sonde (RECAS) for accessing and studying subglacial lakes" or "A prototype of steering RECoverable Autonomous Sonde (RECAS) for accessing and studying subglacial lakes: Design and test".**

Thank you for the suggestion. We have decided to revise the article title to the first option you proposed.

**2.) Line 11 and Line 23: "Thermal sonde", "hot-point" and "thermal drill" have similar meaning, please use the same name throughout the manuscript.**

"Thermal sonde" and "thermal drill" represent distinct concepts with specific meanings. The term "hot-point" has been replaced to " thermal drill" for consistency. Additionally, the keyword "thermal drill" has been added to improve searchability and relevance.

**3.) Line 56, 57 and 58: The melt probe IceMole can also perform directional drilling in ice, can you provide more information about the IceMole's design in thermal head and its drilling performance. (References: Curvilinear melting – A preliminary experimental and numerical study; IceMole: a maneuverable probe for clean in situ analysis and sampling of subsurface ice and subglacial aquatic ecosystems)**

The design and operational principles of the IceMole probe, including the directional drilling approach, differ significantly from those of the RECAS (shorter length of 2 m and less, square cross-section, combined thermomechanical drill head, maximum drilling depth ~25 m, and it can only operate in dry boreholes). Additionally, the mathematical model used for IceMole cannot be directly applied to our thermal drill head design, not only because of completely different geometrical parameters but also because it requires a specific approach to solving the equations by enumerating the parameters. This will require large computing power, which in our case has to be placed inside the sonde, which will reduce its reliability in the long term. To expand the background, we have included a reference to the IceMole project in our article for completeness.

**4.) In the manuscript, there are many names of the thermal head, such as "melting head", "thermal head", "drill head", "melting tip", "thermal drill head", "thermal drill bit", please use the same name throughout the manuscript.**

All terms have been replaced with "thermal head" for text consistency.

**5.) Line 171 and 175: Is there any reference for the formula 5 and 6? How do you get the formulas?**

Formula 5 is sourced from Zvarygin (2010). Formula 6 was derived following the methodology outlined in Shamshev et al. (1983) and has been presented in this manuscript in full (not simplified) form. In Shamshev et al. (1983), only the simplified form of the Formula 6 (Formula 7) is presented directly.

**6.) Line 184: If the radii of curvature ensuring RECAS passibility is 300-600 m, is it means that the RECAS is very difficult to bypass obstacles in the ice unless the RECAS start to deviate before a long distance to the obstacles? If yes, please clarify.**

In general, yes. If an obstacle is encountered, bypassing it by deviating the borehole will require drilling upward for the necessary distance and then manually adjusting the borehole trajectory. Despite the relatively large radius of curvature required, we believe that this approach is feasible for bypassing small obstacles. (At depths of several kilometers in ice, it is unlikely we would need to bypass an obstacle like the size of a 2-3-m diameter stone.)

**7.) Figure 10: What is the function of the limiter?**
The limiter is designed to restrict the position of the pressure chamber during installation within the housing.

**8.) Figure 11: It would be better if the photo can show the full thermal head, such as pressure chamber, thermal sensor and power connector etc.**
The pressure chamber is shown separately in Figure 13. We agree that a photo of the assembled thermal drill head showing the thermal sensor and power connector could be helpful. However, we decided that the specific appearance of the standard factory thermal sensor and power connector look less important for understanding the structure of the thermal drill head, then photo of internal modified parts.

**9.) Line 288: "Dual axis inclinometer" or "two-axis inclinometer"? Please use the same name.**
To avoid repetition and improve readability, the text has been adjusted to "inclinometer with two axes" instead of "two-axis inclinometer."

**10.) Line 311: Polar Research Center laboratory of Jilin University?**
yes.

**11.) Line 334: How much is the 50% power and how much power is increased?**
(Line 269) The total power is approximately 7.6 kW, so 50% corresponds to half of this amount. Specifying the exact value in kilowatts is unnecessary here, as total power may vary.
Regarding "how much power is increased" – thank you for identifying this inconsistency. The power was increased to 100%. The latter part of this sentence and the following sentence have been reordered, to improve understanding and readability. The corrected version of the text now reads:
"Subsequently, the power was increased to the maximum, and half of the heaters on one side of the thermal head were switched off. Then, the ROP was increased to 1 m/h."

**12.) Figure 15: It is better to have depth data in the figure.**
We opted to add the second X-axis with depth data.

**13.) Line 342-344: After the four heater pairs on the opposite side were switched on, the inclination angle decreased to nearly zero and then gradually increased in opposite direction? If so, please clarify.**
No, after the four heater pairs on the opposite side were switched on, the four previously powered heater pairs were switched off. (line 344) You understood the essence correctly. To further clarify this, a corresponding comment has been added to the description of the section of Fig. 15.

**14.) Line 364-373: The two paragraphs should be part of the section of 5.3, which shows how you perform the four experiments. In section 5.2, only the preliminary test procedure and test results should be included.**
We'd like to keep these paragraphs in section 5.2 because these are resulting conclusions of preliminary experiment.

**15.) Line 369: Please present more information about "automatic alignment mode"? How do you control the power?**
To clarify the concept of "automatic alignment mode," Section 2.1 has been expanded with additional details. The other procedures were explained in Sections 2.1 and 2.2.

**16.) Line 434: What kind of RECAS parameters?**
The RECAS parameters include overall dimensions (length, diameter), weight, configuration of the thermal drill bit, number of heaters, diameter, and power consumption.
The parameters for the RECAS-200 and RECAS-500 are described in Table 1, line 81.
The parameters for the testing sonde prototype are provided in lines 261 and 268.

**17.) Line 444: The steering capability of melt probe can be used to maintaining verticality within the desired range or bypass obstacles in the ice. However, according to the research, it looks that a long melt probe is difficult to bypass obstacles in the ice because of its large radii of curvature. In RECAS**

**situation, do you have other methods to bypass obstacles in the ice except for hinged joints?**

Hinged joints will not directly assist in bypassing obstacles. Other methods for obstacle bypassing, aside from those described in response to point 6, are not currently being developed in this project.

**Response to Referee #3**

Thank you for your detailed comments.

Since the reviewer did not number their comments, for simplifying the response process, the text of the comments was divided into 62 points. A reply has been provided for each point.

All references cited in the response comments are relevant only to the original version of the manuscript.

*447 In future work, we plan to conduct experiments on a larger scale (e.g., with a borehole depth*
*448 of approximately 10 m) to refine the results in a deviation intensity range closer to that obtained*
*449 with a real RECAS.*

**1. Please consider submitting a new version of MS after the experiments in the cold well in a borehole at the entire length of the probe model positioned in the borehole.**

At this stage, we conducted experiments with a reduced-length sonde prototype to develop the methodology and confirm that steering is feasible with a RECAS-type sonde. All observed patterns, with appropriate adaptations, can be applied to the full-scale RECAS model in the future. Such experiments are planned, contingent on the availability of funding.

**2. The MS version may be titled "A new technique for thermal ice penetrating probes steering."**

We agree that the title of the manuscript should be revised. We decide changing it to: "A steerable system of RECoverable Autonomous Sonde (RECAS) for accessing and studying subglacial lakes"

**Significant comments and recommendations.**

**3. At a 4-degree angle, the experiment requires deflection of the top of the sonde prototype for about 80 mm. It can be achieved in a borehole of about 300 mm in diameter (80+80+~140; 140 estimated housing OD). The authors did not specify the borehole diameter.**

No, a borehole does not need to be approximately 300 mm in diameter to achieve a 4-degree inclination from vertical. The experiment was planned with the assumption that the borehole had already reached a 4-degree inclination, and our goal was to simply correct its trajectory.

**4. In Fig 14, one can see that the experiment(s) was/were conducted with a partially hung top of the sonde prototype. The feeding rate of the cable is not presented, but data Figs 14-17 depends on it.**

The rate of penetration (ROP) in the RECAS system is controlled via the winch, i.e. feeding rate of the cable is equal to ROP. The ROP values are presented in Fig. 15 as a chart, where this parameter varies over time. For Figs. 16 and 17, the ROP remains constant and is specified in line 365.

**5. Moreover, the demonstrated deflection angle correction can only be achieved with a low cable feeding rate.**

The term "low cable feeding rate" is not clearly defined, making it difficult to determine the specific issue being raised. The experiments were conducted at a drilling speed of 1.5 m/h (line 365).

**6. The presented data must be corrected for cable feeding speed and include angular changes of cable and the probe itself.**

As far as it was possible to understand, the "cable feeding speed" (feeding rate of the cable) has already been discussed in our response to point 4. We agree that considering the angular changes of the cable could be useful, as the cable's angle may affect the system's operation. However, at this stage, it is not feasible during actual drilling to either control the cable's position relative to the borehole or measure its angle relative to the probe. Therefore, we decided to disregard these data in the current project. Additionally, in this experiment, the cable was subjected to a load of only approximately 10 daN. For future testing of a full-scale RECAS model, where the cable load will be significantly higher, we plan to account for the influence of the cable angle. The angular changes of the sonde itself are presented in all the graphs.

**7. The most confusing term used throughout the MS text is PWM coefficients. Example at line 371.**

The term "PWM coefficients" is defined on line 92. Regarding line 371, an explanation is already included in parentheses within the same line.

**Several other undefined parameters are used in the MS, and they must be defined.**

**8. Conversion of Fig 2 to 3d format may allow a better understanding of the theory (more below).**

We believe that a 2D diagram is more than sufficient to graphically represent all geometric calculations based on the inclinometer data and its fixed position relative to the heaters in the thermal drill head. However, we have decided to slightly modify Fig. 2 to improve its clarity.

**9. Rewrite sections 2.2 and 2.3 based on Fig 2 in 3d format.**

As the third dimension is not used in the calculations, we believe there is no reasons to rewrite Sections 2.2 and 2.3.

**10. 2.3. RECAS positioning calculation example – Possibly, authors mean instant inclinometer reading – not an absolute RECAS position. ??? Please clarify. I believe the new Fig 2 in 3d format will help for a clear understanding of the math and meaning of unique terms.**

No, the inclinometer's instant readings do not provide explicit information about which heaters need to be activated or their power consumption limits. The inclinometer only provides deviation data relative to the X and Y axes (line 104).
The response regarding Fig. 2 in a 3D format was provided in point 8.

6) *4.1.1. General testing stand design.*

**11. a) I believe these materials have been published by authors before.**

These materials have not been previously published.

**12. b) In any case, I'm not sure if it deserves to be published. It is a trivia technique commonly used in lab and field applications in the last few decades of the previous century.**

We believe that including a brief description of the test stand is necessary for the readers' convenience and to save their time. This description provides important details for understanding the experiments discussed in the article.

***Specific comments.***

**13. "… the borehole trajectory cannot be controlled." The onboard tilt sensor allows control of the winch feeding rate, making pendulum steering possible. The same can be achieved with the "parachute" heater at the top of a probe. The question is, what option will be more reliable and power efficient?**

We believe that this method does not align with the RECAS concept. The pendulum-steering method appears to be more complex compared to other approaches due to the need for an additional upper heater, which would bear the majority of the probe's weight, and the requirement for a sophisticated automatic control system.
Furthermore, incorporating an upper heater would increase the borehole diameter, leading to higher power consumption and/or reduced rate of penetration (ROP). It would also complicate upward drilling. Therefore, we consider the system proposed in this article to be better suited to the RECAS concept and a simpler solution in terms of implementation.

**14. … to control the directional heat distribution on the drill head surface…**
**15. Consider replacing drawbacks with limitations.**
**16. Consider replacing possibilities with capabilities.**
**17. Consider replacing parameter detection with a monitoring system.**
Corrections 14-17 were accepted.

**18. Consider removing - similar to how a spider climbs on its silk line.**
We would like to keep this analogy, as it provides a clearer explanation of the operating principle.

**19. 79-80. Confusing sentence. Are all heaters powered and not controlled? Consider removing - all heaters.**

Yes, in both prototypes, the heaters are not controlled. Removing "all heaters" would change the meaning of the sentence.

**20. 90-98. A bit confusing sentence due to excessive details. Maybe like that?**

*The sonde is steered using data from an inclinometer installed in the testing probe. The data from the inclinometer are transmitted to a personal onboard computer (PC), processed, and converted into pulse width modulation (PWM) coefficients, which determine the PWM duty cycle for a specified corresponding number of channels heater. In the subsequent tests using the RECAS-200 prototype, the PC will be replaced with a microcontroller mounted inside the sonde. The PMW coefficients are transmitted from the computer to a PWM generator (Sup. 2) inside the sonde prototype, where an individual PWM signal is generated for each channel. Each PWM signal is amplified using a power module (Sup. 3) and supplied to the corresponding heater inside the drill head. The PWM signal duty cycle limits the heater power.*
Since the text was provided without proper markdown formatting, the search for replaced words was conducted manually. Some changes may have been missed.
***in the testing probe*** – To maintain consistency, the term "sonde" is preferred throughout the text. Thus, it was replaced with a pronoun to avoid repetition.
*personal **onboard** computer* – A standard x86-compatible laptop was used in the tests, not a specialized "personal onboard computer".
***Heater*** – No, adding this word makes the meaning unclear. It gives the impression that the number of channels must match the number of heaters, which is not the case. It also creates a false impression that the heaters are connected directly to the PWM signal generator, which is also incorrect.

**21. Another option is to add a block diagram of signals passing through.**
Detailed electrical schematic is shown in Fig. S5.2 (Sup. 5).

*Fig. 1. General schematic of the RECAS-500 with a 500-m-long cable inside (all dimensions are in mm)*

**22. Q – show inclinometer position. If the RICAS stem is flexible, then multiple inclinometers could be shown.**
Inclinometer of RECAS-500 is located in monitoring section. For more detailed information please refer to (Sun et al., 2024)
The body of RECAS-500 is solid and unflexible.

**23. Rotating collector. Likely, it is a slip ring. (Fig. 2)**
Since these terms are synonymous, we prefer to retain the original term "rotating collector," as it was used in the referenced article by Sun et al., 2023.

**24. Fig 2. Please show all parameters used in eq.2 in this figure. Convert Fig 2 in 3d format.**
All parameters used in Eq. 2, except for $\alpha_n$, are shown in Fig. 2. Including all eight $\alpha_n$ angles in the figure would make it visually cluttered and difficult to interpret. We decided to expand the Fig. 2. A diagram illustrating the eight $\alpha_n$ angles is presented as a separate figure labeled as 2B.
Regarding the conversion to a 3D format, this was addressed in point 8.

*For the calculation example, random inclinometer values are taken as: X = 4.5; Y = −3.*

**25. 134, 136, Please define and show in Fig 2 - the projection length of a heater.**
Thank you for your comment, which helped identify an inconsistency in the text. Errors in lines 113, 115, and 134 have been corrected, and overall consistency has been improved.
The projection lengths were already shown in Fig. 2.

**26. 164, 165, 169, ....the deviation intensity – ??? Time or space derivative of R, or angle ??**
Deviation intensity is a commonly used term in directional drilling. Deviation intensity of a borehole is defined as the change in the angle or angles of the borehole over a certain interval along its axis. The unit of deviation intensity is degrees per meter (deg/m).

**27. zenith deviation intensity – what is it? Define the zenith deviation intensity.**
Zenith deviation intensity of a borehole is defined as the change in the zenith angle of the borehole over a certain interval along its axis.

**28. 189-256. I would remove this section. 1) you published it before; 2) These are well-developed devices. In use for many decades.**
Regarding Section 4.1.1 (lines 191–201), a response was already provided in point 12. The following addresses only lines 202–256.
These materials have not been previously published.
Section 4.1.2 (lines 202-224) does not describe devices.
We believe that, as with Section 4.1.1, these sections contain essential information necessary for understanding the experiment described in the article, as well as the operation of the test stand and the system as a whole.
**What for to publish Sup 1…5? All materials are either well-known or commercially available. For instance, for DC-DC converters, see Vicor products.**
Supplementary sections are not reviewed.
*Regarding DC-DC converters. Given the transparency of the review process, we believe that suggesting a specific company without technical justification or clear relevance could be interpreted or perceived by other readers as promoting (advertising) that company's products, which could undermine the integrity of the review process.*

**29. 308 Consider the following change. 5. Laboratory testing of self-steering sonde prototype**
Comment is not clear.

**30. Confusing statement. "ROP, which was controlled by the winch …" Then, the winch feeding rate (FR) will interfere with the heater stirring (HS). Please explain – is it gravity-driven penetration and HS, or is it a combination of HS and PS? More above in Significant comments and recommendations.**
Partly the answer to this question has been provided in point 4 and 6. The meaning of the term "heater stirring (HS)" is unclear. The abbreviation "PS" is not defined.

*WOB, which changed with the ROP and was limited by the sonde prototype weight*".

**31. 1) Statement in conflict with Fig 15 data.**
**It can be so if the winch FR does not follow the sonde prototype ROP (corrected for changing the cable and the sonde itself angle).**
**Also, see below345-349.**
The only way to influence the WOB in this experiment is by adjusting the ROP. The reasons for this are explained in the response to point 4,6 and 30. For more details about the design features of the RECAS full-scale prototypes, please refer to the references.

**32. MS does not present the "purl" steering with heaters. It is FR+HS steering. Please clarify.**
Comment is not clear due to unclear meaning of the term "purl".

**… affected by the controlled heat power INCERT *azimuthal* distribution on the drill head surface, which was controlled by limiting the heater power.**
For better consistency of the text, we decided to insert the word "directional," similar to the approach in point 14.

**33. limiting the heater power.  > controlling**
"Controlling" is a broader term, while in this case, "limiting" is more precise and accurately reflects the intended meaning.

**34. 332 - Fig. 15. Recording of the preliminary experiment. Maybe – Parameters/data of the preliminary experiment.**
We prefer to keep the old name.

**35. 345-349. It stated that the probe prototype (PP) was partly hanged, so the winch feeding rate affects the ROP. ???**
Response was already provided in points 4,6, 30 and 31.

**36. Below Fig 15, I would add applied power and cable feeding rate (not linear, somewhat variable) graphs.**
Detailed information about the power consumption has already been extended (provided as a textual description). Regarding the cable feeding rate, this has been addressed in point 4.

**37. Explain what are "the transients"?**
Transients refer to temporary, fluctuating changes in the system's measured parameters that occur when there are sudden adjustments or disturbances, such as changes in WOB or ROP. These transients create rapid fluctuations affecting the accuracy of feedback in the PID controller. For a PID controller, which relies on stable, predictable input signals to make smooth adjustments, transients are problematic because they introduce noise and instability into the measurements. This can make it difficult for the controller to determine the correct adjustments to WOB or ROP, as it cannot differentiate easily between these temporary fluctuations and meaningful changes in the drilling conditions.

*356-357 up to 373. Hopefully, the revision of the MS section 2.2. and 2.3. allowed me to understand the meaning of this statement.*

**38. Consider replacing the "power consumption" with applied power.**
We would like to keep "power consumption".

**39. "alignment mode" - explain the term.**
To clarify the concept of "automatic alignment mode," Section 2.1 has been expanded with additional details.

**40. … correction coefficient T … . T – is correction coefficient? How does it relate to PWM parameters?**
Please refer to lines 120-126 and Eq. 3 for the explanation of the correction coefficient *T* and its relation to PWM parameters.

**41. The ROP was kept constant at 1.5 m/h. How?**
The ROP was maintained at a constant 1.5 m/h by setting the winch servo drive to operate in constant speed mode. Further details can be found in Sup. 1.

**42. The WOB stabilised between 22 and 28 daN. How? Winch feeding rate?**
It was done empirically through adjustment POR during the preliminary experiment, as described in line 341.
**Then the sonde weight is partially hanged.**
Yes, (addressed in point 4,6, 30 and 31).

**43. The *applied* power consumption was limited to 50% by setting yoff = 0.5.**
We would like to keep this sentence unchanged.

**44. Is the yoff the power coefficient?**
No. Please refer to lines 122-123

**45. Then yoff is proportional or equal to the duty cycle. (?) Why not call it the duty cycle?**
No, $y_{off}$ influences the resulting PWM duty cycles but is neither equal to nor proportional to them. This is because $y_{off}$ is a single coefficient, while there are eight PWM channels (in this experiment) with their respective duty cycle values.

**46. Then what is T concerning the PWM?**
Response was already provided in point 40

**47. The most confusing terms used throughout the MS text are PWM coefficients: T and yoff.**
These coefficients are part of the logistic function (line 118) defined in Eq. 3. To facilitate a better understanding, we provided a calculation example (lines 127–159) with graphs illustrating the effects of these coefficients (Fig. 4 and Fig. 5). Additionally, explanatory text is included in lines 120-126 and 149-

159. For more detailed information on how the logistic function works, please refer to the book by Kyurkchiev et al. (2015).

*It is worth noting that decreasing the PWM **coefficient** recalculation frequency…*

**48. Then, T is the PWM frequency. (?)**
No. Response was already provided in point 40.

**49. Do you control the duty cycle and frequency?**
Yes, the PWM generator allows for adjustment of both the duty cycle, and frequency within a range of 50 to 4000 Hz (see Sup. 2). The PWM frequency is critical because it impacts the stability and longevity of the power modules (line 295) and the heaters.
**High** PWM frequencies increase the thermal load on the MOSFETs in the power modules, causing them to generate more heat. This is especially important since the modules operate in an enclosed space with minimal air circulation.
For the heaters, operating at **low** PWM frequencies in the key mode can mechanically damage the insulator due to thermal cycling. Additionally, in laboratory conditions, where the power supply is the mains network (50 or 60 Hz depending on the country), PWM frequencies close to these values are highly not recommended.
To ensure stable operation under laboratory conditions using a 380V three-phase power supply, a PWM frequency of 903 Hz was selected.

**50. The higher the frequency, the more sensitive the control is.**
If this refers to the PWM frequency, the answer is no. The PWM frequency has minimal impact on control sensitivity due to the significant thermal inertia of the thermal head.

**51. Then, set the high PWM frequency why it has to be controlled.**
If the comment refers to PWM frequency, the explanation was provided in point 49.

**52. The big mass of a heater does not require high-frequency control.**
The term "high-frequency" is not quantitatively defined in this comment, making it difficult to provide a specific response.

*"It is worth noting that decreasing the PWM coefficient recalculation frequency (i.e., slowing the response to inclination angle changes) can influence the borehole deviation intensity."*

**53. Please explain why PWM frequency needs to be controlled.**
It is unclear which frequency is being referred to in the comment.
If the comment refers to PWM frequency, the explanation was provided in point 49.
If the comment concerns the PWM coefficient recalculation frequency, this was addressed in point 7 and is further explained in the quoted statement: "can influence the borehole deviation intensity" (line 371).
*Additional clarifications:*
*The recalculation frequency is not actively controlled in this experiment. Instead, it remains constant at 1 Hz (line 370). However, it is suggested in lines 372–373 that reducing this frequency most likely could impact the results. Therefore, while the recalculation frequency was fixed in these experiments, its value should be at least accounted for and potentially adjusted for future experiments.*

**54. Instead of determining the optimum frequency for the specific device?**
It is unclear which frequency is being referred to in the comment.
If the comment refers to PWM frequency, the explanation was provided in point 49.
If the comment concerns the PWM coefficient recalculation frequency, the explanation was provided in points 7 and 49

**55. 374-424.  5.3. Experimental results and analysis… Discussion of ….?**
Correction was accepted

**56. Possibly, the extended version of Figure 18 with R=infinity (strate borehole) and sections 1.0°-1.5° and 2.5°-4.0° allow for easy comprehension of the idea.**

As Fig. 18 is schematic, it is designed to illustrate an example calculation for a single section. All other sections are calculated using the same methodology, and duplicating similar figures not provide additional value.

**57. How does the time scale in Fig 16 correspond to the time scale in Fig 15?**
Since Fig. 15 and Fig. 16 pertain to different experiments, their time scales correspond only in that both are measured using the same High Precision Event Timer (HPET).

**58. .. "*demonstrates that correction coefficient T …*" Is it the same T as before or another one?**
There is only one variable denoted by the letter "*T*" throughout the entire article, and it is consistently referred to as the "correction coefficient *T*."

**59. Please define - the automatic alignment length.**
Response was already provided in point 39

431-432. *The borehole* **deviation intensity** *during drilling can be corrected by controlling the correction coefficient T.*

**60. Is it the same T as above?**
Yes

**61. … experimental drilling parameters, the borehole deviation radius intensity must be reduced.**
Correction was accepted.

**62. 437-442. Sequencing of RICAS could be a dangerous strategy for an expensive field program.**
The use of RECAS involves risks inherent to any drilling program. The most significant risk in RECAS is the failure of the upper thermal head and side heaters. Therefore, during assembly, special attention will be paid to ensuring the reliability and durability of these systems.

443 … *Sonde passability at large borehole deviation intensity values can …*

**Response to Referee #4**
Thank you for your detailed and valuable comments and questions.
All references cited in the response comments are relevant only to the original version of the manuscript.

**General comments/questions that lack explanation:**

**- How do you avoid rotation along RECAS axis, which would change the direction of the steering?**
We do not avoid rotation along the RECAS axis. Since the inclinometer is rigidly attached to the thermal head, any rotation along this axis does not affect the direction of steering. The system automatically calculates new PWM coef. for the heaters based on the updated inclinometer readings.
To improve understanding of this aspect, the description of the inclinometer (lines 288–291) has been expanded.

**- Why do you use the non standard unit daN instead of kg for weight specifications? I recommend conversion throughout the paper.**
We partially agree with this comment and have replaced the unit daN with kg in Table 5 and line 347. In other instances, we have retained the unit daN as it describes weight as a force rather than mass.

**- You have not adequately proven that this method would work in pactise in the field, with a 7 meter RECAS fully submerged. A field test would be best; however expensive.**
**At least provide better, more inclusive calculations to show the feasability.**
The experiment was designed based on general principles of borehole deviation, which are independent of the sonde's specific parameters. The experimental parameters were chosen to ensure that, within the 0.5 m borehole segment, sufficient data could be obtained to validate the Eq. 3, analyze its behavior, and provide a proof of concept for the auto-alignment mode.
There are two potential ways to apply the proposed method to a full-scale prototype: **Reducing Borehole Deviation Intensity**, that can be achieved by limiting the angle or adjusting the auto-alignment

mode parameters during its operation; **Improving Sonde Passability** (lines 437–446).
Alternatively, a combination of both approaches could be utilized in practice.

**Given the icesheet thickness at the glacial lakes positions, how much maneuvability is needed and how much does your method provide?**
The main capabilities for maneuverability and their calculations are described in Section 3.
The required level of maneuverability will depend on the specifications of future RECAS models. For the RECAS-500 prototype, an approximate calculation was provided in Section 3.
As for how much maneuverability our method provides, the method can be adapted to a certain extent for any RECAS configuration. Primarily, we rely on the system's ability to maintain the sonde in a vertical position during drilling, resorting to borehole deviation only in cases of insurmountable obstacles.

***Comments and questions:***

**line 26-28: It is widely accepted that there are subglacial lakes as also shown with references to Bowling and Siegert. But you also mention rivers and streams. Do you have references for this?**
A reference (Ashmore et al., 2014) has been added.
doi.org/10.1017/S0954102014000546

**line 31: Do you have a reference for a glacial lake being several hundred metres deep?**
A reference (Wright et al., 2011) has been added.
doi.org/10.1029/2010GM000933

**line 47: To provide a fair comparison; please put numbers instead of adjectives when describing the other technologies and then give the same numbers for RECAS for comparison.**
**line 52-53: You mention the cost and required specialist staff. Please provide the numbers for both hot-water drilling and RECAS for easy, quantitative comparison.**
Thank you for pointing out these inaccuracies. The corresponding paragraph in the introduction has been expanded. Additionally, a reference (Siegert et al., 2012b) has been added.
doi.org/10.1029/2011RG000361

**line 119, equation 3: What is the reason for this specific notation being used? Why not use exp(ln(a))=a and exp(a+b)=exp(a)*exp(b) to simplify?**
You are right, the final equation can be simplified and eliminated the natural logarithm.
The equation will look like this

$$K_n = \frac{y_{off}\exp\left(Tl_n\right)}{y_{off}\exp\left(Tl_n\right) - y_{off} + 1}$$ or like this $$K_n = \frac{y_{off} - 1}{y_{off}\exp\left(Tl_n\right) - y_{off} + 1} + 1$$

However, because the intermediate coefficient V will be used in future calculations and experiments, we chose to retain the original form to ensure the ability to reference this paper directly.

**line 180-181: please detail how the clearance looks along the 7 meters of length, as this influences the achievable control.**
Clearance is highly correlated with ROP. For more details, please refer to Li et al. (2020). At a constant ROP, the clearance remained stable along the 7 meters of length.
**How is the melt water pattern surrounding the head?**
During drilling, only a thin layer of water forms between the surface of the thermal head and the ice. This question has been addressed in numerous theoretical publications by researchers from St. Petersburg Mining Institute and Kazan University (unfortunately, these are available only in Russian). We recommend the monograph: Chistyakov, V.K., Salamatin, A.N., Fomin, S.A., Chugunov, V.A. *Heat and Mass Transfer in Contact Melting.* Kazan, Kazan University Press, 1984, 176 pages.
**How does the continous reclosing of the ice affect the deviation intensity?**
Based on the experiments described in Wang et al. (2024) (doi.org/10.3390/w16233460), meltwater

refreezing begins approximately 7.3 hours after drilling. Therefore, we believe it should not have any significant effect on deviation intensity.

**line 198 table 5: What is "ROP"? What is "tripping speed"?**
ROP – rate of penetration (line 181)
tripping speed - the speed of lowering and lifting drilling equipment, usually inside the borehole.

**line 204: Did you check the measuring wheel for slippage?**
The wheel shaft is mounted on ball bearings, and it was experimentally verified that, within the range of ROP used in the experiment, slippage of the Kevlar rope relative to the groove of the aluminum wheel can occur only if the tension on the rope is low (i.e., the weight is less than 100 grams).

**line 208: you mention a top wheel, but this is not mentioned in figure 7.**
Thank you for pointing out this inconsistency. The term "top wheel" in line 208 has been replaced with "measuring wheel assembly" in the text for consistency.

**line 210: accuracy is the wrong term. You mean resolution. Please correct this throughout the paper in several places.**
Thank you for pointing this out. The term "accuracy" has been replaced with "resolution" in lines 210, 212, and 218.

**line 214 equation 8: Do you not need to divide d by 2 ?**
not need.

**line 222 equation 11: You have assumed d=0. This is not mentioned or reasoned for.**
Thank you for pointing out this inaccuracy. Eq. 10 contains an incorrect definition. "Wheel diameter" has been replaced with "diameter of the rope axis on the wheel", and accordingly, for l, "wheel circumference" has been replaced with "rope length passing through the wheel per revolution".

**line 244 table 6: Are these the in-situ accuracies or merely assumed based on the datasheet? I.e. how is the thermocouples thermally anchored and isolated to ensure this accuracy?**
The accuracy information was taken directly from the datasheet.
During the experiment, we measured the air temperature, ice temperature, and the temperature inside the control box. These data were used solely for monitoring purposes.

**line 244 table 6: Also are the percentages for the accuracy given as %reading or %fullscale?**
For the thermocouples and load cell, the accuracy is given as %reading. For the voltage and current sensors, the term "准确度" is used in the datasheet, which most likely also refers to %reading.

**line 274 figure 11: The thermal sensor is placed upper left - why here and what is it measuring?**
Temperature sensor is used to monitor the thermal head temperature. For more detailed information please refer to (Sun et al., 2024)

**line 305 table 7: You mention the angle accuracy is 0.2°. What is the reference? What was the project requirement? Is it adequate to be able to stay within the 300-600m turning radius?**
The accuracy information was taken directly from the datasheet. For this experiment, such accuracy is acceptable. In the full-scale RECAS prototype, a higher class and different type of position sensor will be used.

**line 313: How is the ice temperature kept constant?**
The ice blocks are stored in a refrigerated chamber at a constant temperature of -16°C. For the experiment, a block was removed from the chamber, and an experimental run was conducted within 40 minutes. Before the experiment began, the temperature of the outer ice layer changed by less than 1°C.

**line 324 figure 14: Please provide a size reference in the picture.**
A size reference has been added to Fig. 14.

**line 331 figure 15: Please widen the graph for better readability.**
Fig. 15 has been enlarged to the maximum page width.

**Also please add below the graph the power and what heaters are on/off for better readability.**
**line 334: Please provide a number for what is the power increased to?**
**line 335: When on the graph are half the heaters switched off?**
**line 337: What is the power for 2 m/h?**
To improve understanding and readability the textual description of fig. 15 (333-344) has already been extended.

**line 441-442: Please provide a calculation example how a few millimeters would affect passability.**
We believe that including such calculations in the text of the article is unnecessary, as the relevant equations (Eq. 6 or Eq. 7) are already provided. However, as an example here, a 5 mm change in our case would reduce the range of radii of curvature from 300–600 m to approximately 250–400 m.

***Comments and questions for supplementary material:***

**S1, equation S1.3: The units do not fit together.**
Eq. S1.3 is derived from Eq. S1.1. Eq. S1.1 was taken directly from the operation manual. It is likely that the equation is presented in a simplified form, where the coefficients having specific units.

**S1, page 4: Please provide the torque-velocity graph for the servo drive.**
The torque-velocity (speed) relationship for the servo drive in nominal operation mode (0–3000 rpm) is constant and equal to 0.64 Nm.

**S3, page 1: You mention decoupling capacitors twice, but referencing the difference capacitors on different figures. Please reference figure S3.1 for the 2.2 uF capacitor and fig S3.2 on the second last line on the page.**
These capacitors serve different purposes:
fig. S3.1 - 2.2 uF capacitor used to decouple from high-frequency noise
fig. S3.2 – two 100 uF capacitors used to prevent leaking low-frequency switching noise of PWM controlled heaters
Their purposes are described in the text.

---

## Referee Report (RR1)

This paper outlines the general principles of steering RECAS and discusses experimental results of deviational ice drilling with a controllable electric thermal head. This is a very interesting study, and the quality of the manuscript has been greatly improved after revisions. However, there are some formatting issues that need to be noted, such as table titles and diacritical marks being on the same line; The font in the picture should be as consistent as possible.